# Behavior of Traffic Congestion and Public Transport in Eight Large Cities in Latin America during the COVID-19 Pandemic

**Renato Andara** [1], **Jesús Ortego-Osa** [2], **Melva Inés Gómez-Caicedo** [3], **Rodrigo Ramírez-Pisco** [4,5], **Luis Manuel Navas-Gracia** [2,*], **Carmen Luisa Vásquez** [1] **and Mercedes Gaitán-Angulo** [6]

1   Directorate of Research and Graduate Studies, Universidad Nacional Experimental Politécnica Antonio José de Sucre, Barquisimeto 3001, Venezuela; randara@unexpo.edu.ve (R.A.); vasquez@unexpo.edu.ve (C.L.V.)
2   Universidad de Valladolid, 47002 Valladolid, Spain; jesus.ortego@uva.es
3   Fundación Universitaria Los Libertadores, Bogotá 16-63A, Colombia; migomezc@libertadores.edu.co
4   Universitat Carlemany, 4000 Sant Julià de Lòria, Andorra; rramirez@universitatcarlemany.com or rodrigo.ramirez.pisco@upc.edu
5   Universitat Politècnica de Catalunya, 08034 Barcelona, Spain
6   Fundación Universitaria Konrad Lorenz, Bogotá 62-43, Colombia; mercedes.gaitana@konradlorenz.edu.co
*   Correspondence: luismanuel.navas@uva.es

**Abstract:** This comparative study analyzes the impact of the COVID-19 pandemic on motorized mobility in eight large cities of five Latin American countries. Public institutions and private organizations have made public data available for a better understanding of the contagion process of the pandemic, its impact, and the effectiveness of the implemented health control measures. In this research, data from the IDB Invest Dashboard were used for traffic congestion as well as data from the Moovit© public transport platform. For the daily cases of COVID-19 contagion, those published by Johns Hopkins Hospital University were used. The analysis period corresponds from 9 March to 30 September 2020, approximately seven months. For each city, a descriptive statistical analysis of the loss and subsequent recovery of motorized mobility was carried out, evaluated in terms of traffic congestion and urban transport through the corresponding regression models. The recovery of traffic congestion occurs earlier and faster than that of urban transport since the latter depends on the control measures imposed in each city. Public transportation does not appear to have been a determining factor in the spread of the pandemic in Latin American cities.

**Keywords:** traffic congestion; public urban transport; daily infections by COVID-19

## 1. Introduction

At the end of 2019, a new coronavirus, identified as SARS-CoV-2, was detected in Wuhan, China, which causes a disease called COVID-19 [1]. On 9 March 2020 in Italy, the second outbreak of the virus was detected, leading to mobility restrictions, school closures and measures such as social distancing [2]. The virus quickly spread to other cities and regions in Asia, Europe, Africa, North America and Latin America.

On 11 March 2020, the World Health Organization declared this virus as a pandemic, leading governments to take measures to promote changes in mobility, the way people work and social relations [3]. However, as a result of these actions, there have been closures of businesses and offices, the prohibition of travel that is not strictly necessary, and even a compulsory quarantine at home has been imposed [4]. Its rapid spread led to congestion in health systems and the introduction of restrictions on people to slow down the rate of infection and death. This brought limitations to economic activities and a contraction of the gross domestic product (GDP) of the different economies of the countries [5].

The use of public transport around the world was significantly reduced due to the need to safeguard health and avoid an increase in the number of infections, leading to growth in the use of private vehicles as a means of transport [6]. Indeed, the pandemic

has harmed mass transport, such as buses and subways, as users decided to use private means of transport, such as bicycles and other soft means of transport, adapting routes for these means of transport and walking [7–9] in order to reduce the spread of the virus. It is, therefore, relevant to analyze the impact of the measures adopted by governments and the consequences for the actors involved in the sector [10]. In addition, this pandemic significantly affected the development of higher education and included changes in the way of teaching (the rise of e-learning), especially in the area of student mobility (national and international) due to travel restrictions, campus closures and the need to maintain health and safety [11,12]. Additionally, the restrictions imposed by the COVID-19 pandemic [13] had a greater impact on mobility in low-income groups, compared to high-income groups [14].

The characteristic of COVID-19 has been analyzed by the international community based on reports of new cases as the pandemic has progressed. It is emphasized that the route of infection is by air and by contact with infected people or surfaces that have been in contact with them. Its mortality rate is not high (approximately 2–3%); however, its rapid spread encourages the implementation of protocols to stop its spread [15]. Within this framework, among the generalized measures to prevent the spread are mobility restrictions, such as the closure of borders and airports and, among others, suspension of flights and access to public and private urban transport systems [16]. However, the uncertainty of the forms of contagion and effectiveness of these measures persists [17,18].

The appearance of this disease is a serious alert for society and a challenge that transcends borders, with an effort that requires understanding and rationalizing of the reach and potential of such a threat. However, just as there are doubts about the pharmacological therapeutic measures required, there are also doubts regarding the adequacy and effectiveness of the control measures [18]. Due to the initial COVID-19 outbreak, mass transportation systems in China were closed, resulting in at least 40 million people stopping their travel plans, the economic and social costs of which escape the epidemiological estimates that have been done [19,20]. Similar to this situation, the economic consequences of the pandemic have proven to be very negative, as different levels of mobility restrictions have been implemented in response to the spread of the virus, with the transport of people and goods being among the most affected sectors [5,16].

At the international level, initiatives for infection treatment and public health measures have focused on strengthening non-pharmaceutical interventions, including intensive contact follow-up, the quarantine of individuals potentially exposed to infection, and the isolation of those infected or suspiciously asymptomatic [15]. The potential effects deserve coordinated, timely, and effective actions to prevent additional cases or worse health outcomes [21].

One of the regions that are later to suffer from COVID-19 is Latin America, where the first cases of infection and death from this cause occurred on 26 February [22] and 27 March 2020 [23], in Brazil and Argentina respectively. The control measures to curb infections in the region are not homogeneous [24], but all of them have directly impacted the mobility of people, among other effects [25–28].

The mobility of people is a social expression, giving rise to patterns of behavior due to different social, cultural, and economic experiences. This guarantees communication at different distances, affecting the regional and world economy. It is constantly changing, is part of urban life, and has an important cultural and political, and economic development component [29].

In the last decades, there has been an accelerated and uncontrolled urban growth in Latin America, producing a great increase in the need for services, among which include mobility and transport. Traffic congestion problems are evident, which has accelerated the need for higher levels of investment in urban infrastructure and transport modalities [29–31]. Currently, due to COVID-19, there is a decrease in the levels of congestion and use of urban transport in the region.

The purpose of this article is to analyze traffic congestion and the use of public urban transport concerning daily COVID-19 infections in eight of the main cities in Latin

America, belonging to five countries, for the period from 9 March to 30 September 2020, corresponding, therefore, to approximately seven months. Descriptive statistical techniques have been used to analyze the relationship between pandemic-associated infections and people's mobility from the point of view of individual mobility, which affects traffic congestion, as well as the relationship with the use of urban transport. The article is restricted to Ibero-America where the cities were chosen because they are the capital cities of their respective countries, because they account for a high proportion of the population of their countries, as shown in Table 1, and because they have consistently high levels of congestion. Additionally, except for Santiago de Chile, all of the selected cities are among the largest cities in the world in terms of the number of inhabitants in 2015 [32]. Figure 1 shows the geographic location of the cities and countries selected in this article.

**Table 1.** Population of the cities and countries under study.

| City (Country) | Population (Number of Inhabitants) of City (Country) | Percentage of City's Population in Relation to Country (%) |
|---|---|---|
| Bogotá (Colombia) | 8,770,058 (50,882,884) | 17.24 |
| Buenos Aires (Argentina) | 14,338,718 (45,195,777) | 31.73 |
| Mexico City (Mexico) | 22,381,714 (128,932,753) | 17.36 |
| Santiago (Chile) | 5,688,218 (19,116,209) | 29.76 |
| Lima (Perú) | 9,609,692 (32,971,846) | 29.15 |
| Brasilia (Brazil) | 2,043,811 (212,559,409) | 0.96 |
| São Paulo (Brazil) | 21,001,688 (212,559,409) | 9.88 |
| Rio de Janeiro (Brazil) | 10,523,151 (212,559,409) | 4.95 |

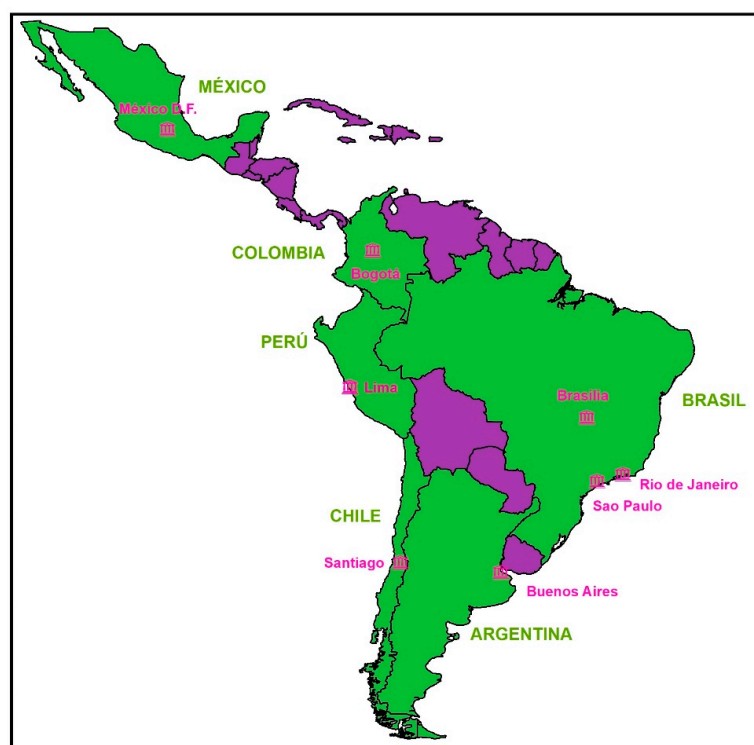

**Figure 1.** Cities and countries (in green) that are the subject of this article.

For the data, the information available on specific official web pages and public social networks was used. Different institutions and organizations have made the data available to the health authorities, the scientific community, and interested parties to understand better the contagion process of the pandemic, its impacts, and the effectiveness of the health control measures implemented. All of this was done in order to disseminate information on the progress of the infection and the basic measures of care, control of travelers, and local plans

to deal with possible cases. For the case of this article, the traffic congestion data published by the Inter-American Development Bank (IDB) [24] were used and for urban transport, the data from the Moovit© application were used [33]. In addition, the information from Johns Hopkins Hospital University [34] was used for the number of daily infections by COVID-19.

The cities analyzed were Bogotá (Colombia), Buenos Aires (Argentina), Mexico City (Mexico), Santiago (Chile), Lima (Perú) and Brasilia, São Paulo and Rio de Janeiro, with the latter three in Brazil. These cities are distinguished by having the largest population in the region [35], levels of congestion [36] and number of COVID-19 infections [37]. Among the public urban transport systems, the metro, bus rapid transit (BRT), surface trains, trams, and trolleybuses [38,39] stand out. Table 2 shows for each of these cities the population density in the metropolitan area, the percentage of modal participation of public urban transport, and the approximate number of daily passengers served in the BRT and subway systems, which are the ones with the highest participation [40,41].

**Table 2.** Data on population density in the metropolitan area, the modal share of urban public transport and the number of daily passengers served in the BRT and subway systems of the cities under study.

| City | Population Density (Inhab/km$^2$) | Modal Share of Urban Transport (%) | Daily Passengers in the BRT | Daily Passengers in the Subway |
|---|---|---|---|---|
| Bogotá | 3347.4 | 59.0 | 2,192,009 | - |
| Buenos Aires | 50.7 | 44.7 | 1,419,000 | 1,500,000 |
| Mexico City | 2450.7 | 77.9 | 1,240,000 | 1,600,000 |
| Santiago | 462.0 | 29.1 | 340,800 | 1,975,000 |
| Lima | 3008.8 | 62.0 | 704,803 | 550,000 |
| Brasilia | 491.6 | 36.2 | 51,000 | 43,800 |
| São Paulo | 2714.4 | 36.8 | 3,300,000 | 4,600,000 |
| Rio de Janeiro | 2208.8 | 48.7 | 3,535,466 | 780,000 |

The metro of the city of Buenos Aires (SUBTE) is one of the first implemented in the region and the fleet of electric BRT of the city of Santiago is the second largest in the world in 2020, which shows that accelerated and giant steps have been taken for the development of public urban transport, in this way, guaranteeing sustainable urban mobility.

Additionally, [42], derived from ICSC-CITIES 2020 Congress, shows the impact of the COVID-19 pandemic on traffic congestion in 13 Latin American cities during the first five months of the pandemic declaration. The results are analyzed by grouping the countries into four conglomerates, showing that as social distancing measures are relaxed, there is a recovery of traffic congestion due to the use of private vehicles. In contrast to [42], the present work includes mobility with private vehicles and urban public transport in its different forms, in eight cities. This allows for a broader view by including all means of motorized mobility for a longer period (seven months), allowing us to analyze people's behavior based on their motorized mobility in a time window where a recovery in the number of infections is beginning to be observed.

## 2. Methodology

The data used in this article were obtained from the IDB Coronavirus Impact Dashboard [24], according to the methodological note [22], in accordance with the agreement of this institution with the Waze© platform [43]. These data come from the information of the mobile phones of its users, which are aggregated and geocoded in real-time every 2 min. The Coronavirus Impact Dashboard uses the week of 1 to 7 March 2020, as a reference for comparison, because traffic patterns were not affected by regional holidays and there were still very few reported cases of infections in the region. Additionally, governments have not yet issued restrictions or recommendations for social distancing.

The IDB Coronavirus Impact Dashboard methodology for creating urban centers as adjacent grid groups limits publication to urban centers with more than 750,000 inhabitants. In other words, it restricts the analysis to centers with sufficient Waze© activity and

historical information, leaving only 64 metropolitan areas in 19 countries, using the Traffic Congestion Intensity indicator (TCI) as an indicator for the analysis, which is the sum of total times and lengths of congestion for periods of 24 h, compared with those carried out on the days of 1 to 7 March 2020 (ΔTCI). This indicator allows the percentage comparison between the same days of the current week, called ratio-20.

The IDB Coronavirus Impact Dashboard [24] publishes information from 64 metropolitan areas; however, for the present study, only the cities of Bogotá (Colombia), Buenos Aires (Argentina), Mexico City (Mexico), Santiago (Chile), Lima (Perú), and Brasilia, São Paulo and Rio de Janeiro, (Brazil) were considered because they are large cities where the impact of public transport is very important. The analysis period ranges from 9 March to 30 September of 2020, approximately 7 months.

In this work, the graphs of variation of the TCI (ΔTCI) and the percentage of reduction of public urban transport are analyzed as a function of time and, in turn, of the cases of infections detected daily for the same days. The contagion data are taken directly from those published by the Coronavirus Research Center of the Johns Hopkins Hospital University [34], corresponding to all countries.

For each area, graphs were made, and descriptive statistics were estimated, based on observing the recovery of traffic congestion and public urban transport as a function of the decrease in infections. These graphs determine the minimum vehicle mobility and, from this moment, the mobility recovery rate (MRR), as an indicator of the recovery rate of traffic congestion in the area analyzed through regression analysis of the data.

To perform the analysis of urban mobility in public transport during the COVID-19 pandemic, we used data collected by the Inter-American Development Bank (IDB) in its Coronavirus Impact Dashboard [24]. This dashboard is based on data from different sources on public transportation. For example, for the city of Bogotá, the information from the BRT Transmilenio open data website was used. For Lima, the information from the PROTRANSPORTE Metropolitan Institute of Lima of the Municipality of Lima was used. In both cases, the daily data were compared with the validations in the week of 2 to 8 March 2020. For São Paulo, the data of the Municipal Secretariado de Mobilidade e Transportes of the Municipality of São Paulo were used and compared with the validations from the week of 15 January 2020. For the rest, the data from the Moovit© public transport application were used and compared with the week of 2 to 8 March 2020.

## 3. Results

Figures 2–9 show the percentages of reduction of traffic congestion and of the use of urban transport, along with daily infections by COVID-19 for the cities of Bogotá, Buenos Aires, Mexico City, Santiago, Lima, Brasilia, São Paulo and Rio de Janeiro, respectively. The analysis period corresponds from 9 March to 30 September of 2020, approximately 7 months.

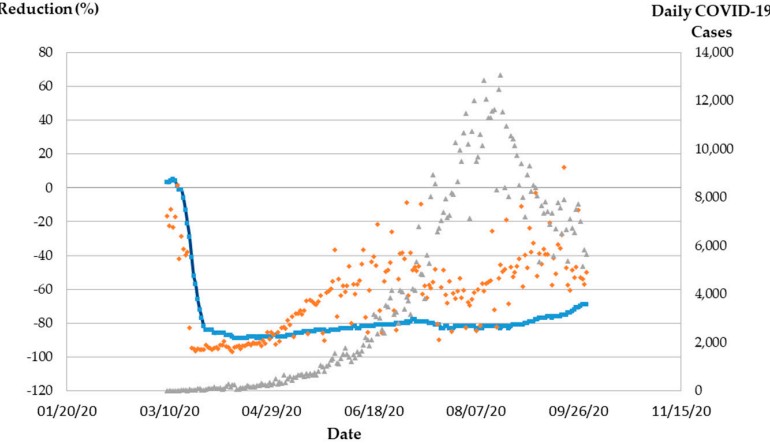

**Figure 2.** Percentage reduction (%) in traffic congestion (red) and urban transport use (blue), compared to the number of daily COVID-19 infections (gray). Bogota city, Colombia.

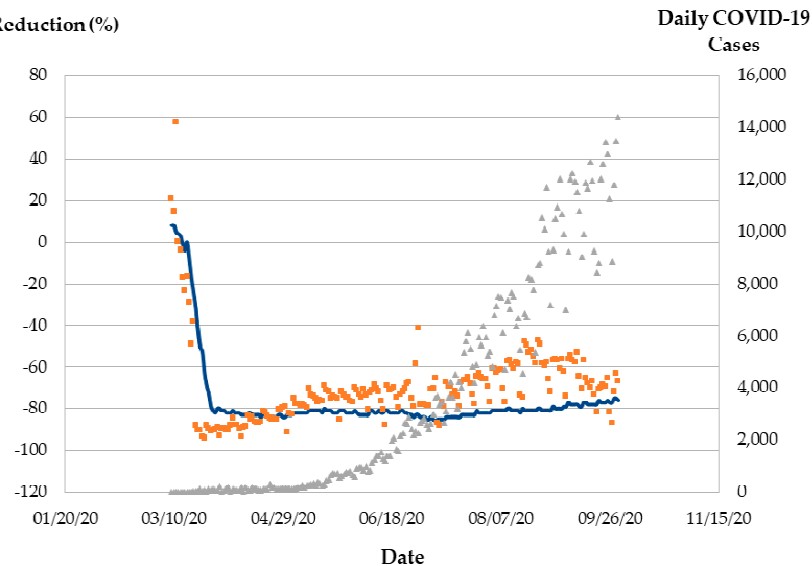

**Figure 3.** Percentage reduction (%) in traffic congestion (red) and urban transport use (blue), compared to the number of daily COVID-19 infections (gray). Buenos Aires, Argentina.

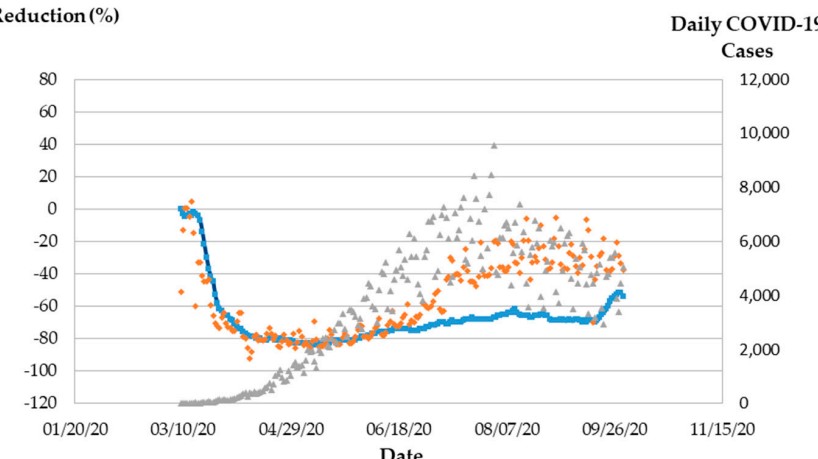

**Figure 4.** Percentage reduction (%) in traffic congestion (red) and urban transport use (blue), compared to the number of daily COVID-19 infections (gray). Mexico City, Mexico.

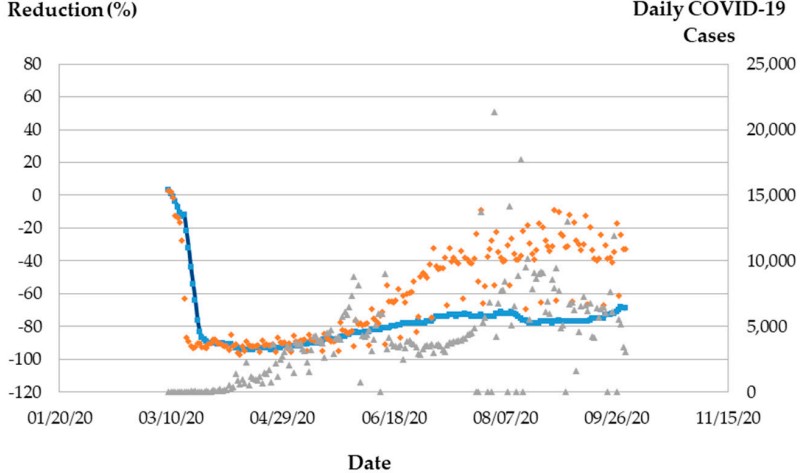

**Figure 5.** Percentage reduction (%) in traffic congestion (red) and urban transport use (blue), compared to the number of daily COVID-19 infections (gray). Santiago, Chile.

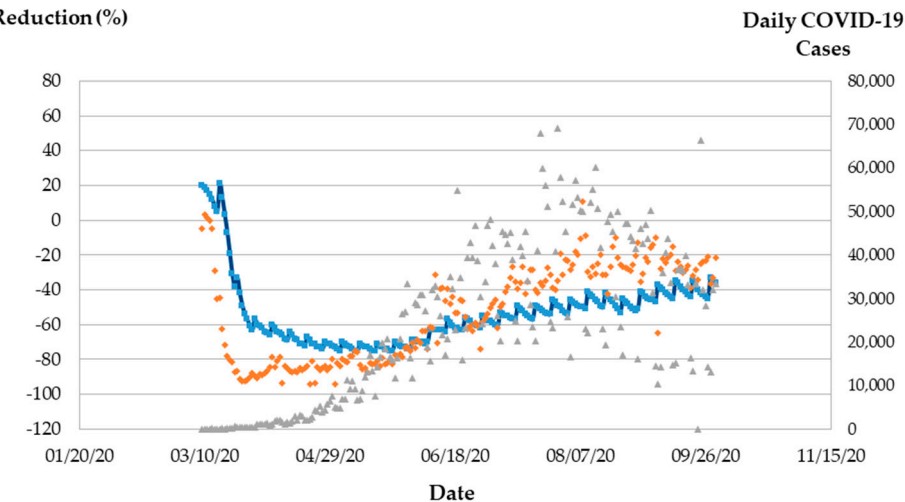

**Figure 6.** Percentage reduction (%) in traffic congestion (red) and urban transport use (blue), compared to the number of daily COVID-19 infections (gray). Lima, Perú.

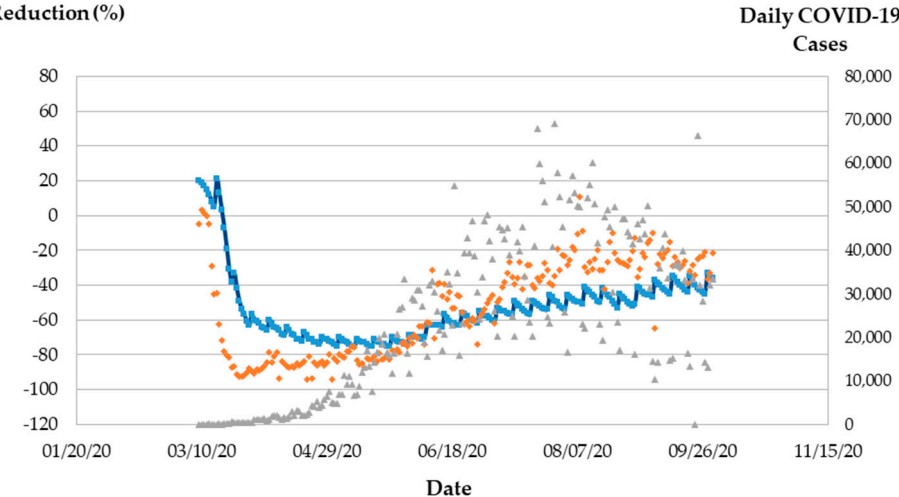

**Figure 7.** Percentage reduction (%) in traffic congestion (red) and urban transport use (blue), compared to the number of daily COVID-19 infections (gray). Brasilia, Brazil.

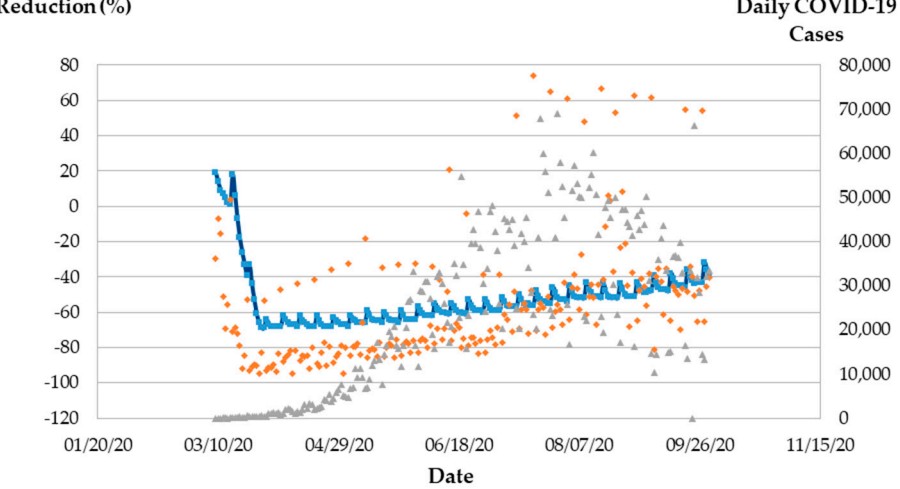

**Figure 8.** Percentage reduction (%) in traffic congestion (red) and urban transport use (blue), compared to the number of daily COVID-19 infections (gray). Sao Paulo, Brazil.

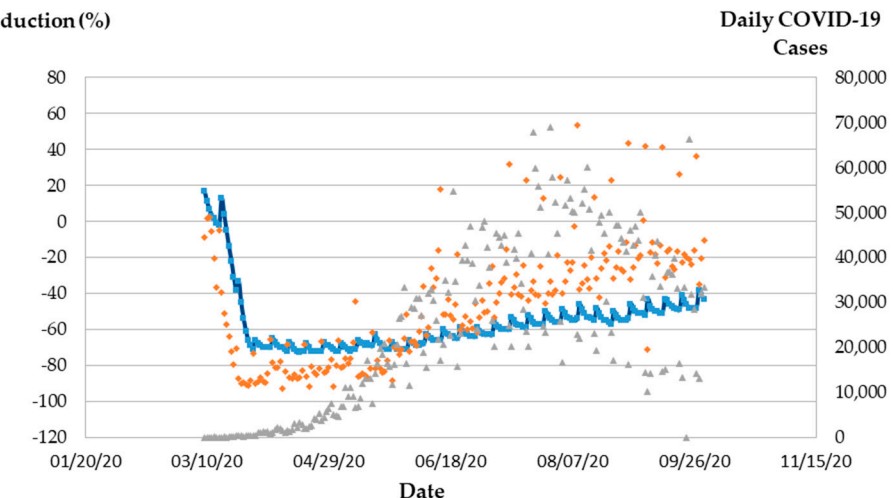

**Figure 9.** Percentage reduction (%) in traffic congestion (red) and urban transport use (blue), compared to the number of daily COVID-19 infections (gray). Rio de Janeiro, Brazil.

## 4. Discussion

Urban mobility is a key element for the economic and social development of cities, linked to the means of transport used (motorized or not) and to a set of externalities, including traffic congestion, correlated with the use of private vehicles and urban transport systems. In this context, urban transport is essential to ensure the social and economic well-being of cities, and the model that articulates order and fluidity and guarantees the comfort, saved time , economy and mobility of its inhabitants is, therefore, of very relevant importance.

Prior to the COVID-19 pandemic, Latin American cities were among the most congested in the world. According to the INRIX© ranking published in 2019 [44], cities such as Bogotá, Rio de Janeiro, Mexico City and São Paulo were ranked 1, 2, 3 and 5, respectively, as the most congested in a universe of 945 cities worldwide; in these cities, a driver loses more than 145 h a year sitting in his or her vehicle and the average speed is less than 21 km/h. However, in the publication of this ranking for the year 2020 [36], there is evidence of a decline in the rankings and hours lost, with driving hours decreasing by 31%, 69% and 66% in the cities of Bogotá, Mexico City and São Paulo, respectively, as a result of the COVID-19 pandemic.

Similarly, in [45] and for the year 2020, traffic congestion drops were reported, comparatively, with respect to 2019, in the cities of Bogotá, Mexico City, Rio de Janeiro, Santiago, São Paulo, Brasilia and Buenos Aires with 19%, 16%, 14%, 13%, 15%, 15% and 11%, respectively. As an example, government restrictions in Colombia reduced demand on transport systems [46] and eventually changed the modes of transport used by the population, in efforts to try to maintain social distance. Many people decided not to use public transport and replaced it with private or non-motorized vehicles, such as walking or cycling [13]. Additionally, the total number of taxi trips in the areas studied in China declined sharply during the pandemic. This decline was most significant for taxi use during night-time hours, with no variation in average trip distances.

The challenges of urban mobility in Latin American cities prior to the COVID-19 pan-demic were as follows [47]:

- Public transport is the most used mode of transport (buses and micro-buses, with 102 million trips per day, followed by metros and trains, with 19 million trips per day), accounting for 42% of trips in metropolitan areas. However, public transport is of poor quality and travel time and cost for users are high;
- Road safety affects the most vulnerable (pedestrians) who account for more than half of all traffic fatalities (10,000 deaths per year);

- The level of pollutant emissions due to transport is very high in cities (260,000 tonnes of $CO_2$ and 3600 tonnes of other pollutants, such as CO and NOx), damaging public health;
- Traffic management is very limited, which prevents the optimization of the existing road infrastructure. Effective priority for buses, pedestrians and cyclists is very low.

The figures previously introduced (Figures 2–9) show that the mobility restriction measures in the first months of contagion achieved a significant reduction in traffic congestion, being the lowest in the city of Lima with 87%. In turn, urban transport falls close to 90% for this same city. Table 3 shows a summary of the variables analyzed for all the cities considered.

**Table 3.** Percentages of maximum decline and mobility recovery rate applied to traffic congestion and urban public transport in the Latin American cities under study.

| City | Traffic Congestion | | Public Urban Transport | |
|---|---|---|---|---|
| | Maximum Decay (%) | Mobility Recovery Rate (%) | Maximum Decay (%) | Mobility Recovery Rate (%) |
| Bogotá | −97.00 | 27.9 | −89.00 | 7.31 |
| Buenos Aires | −94.08 | 14.4 | −86.00 | 9.79 |
| México City | −92.05 | 41.8 | −84.00 | 12.2 |
| Santiago | −92.52 | 44.2 | −88.00 | 17.3 |
| Lima | −97.24 | 45.2 | −94.00 | 11.9 |
| Brasilia | −95.04 | 35.9 | −69.00 | 14.9 |
| São Paulo | −92.74 | 50.4 | −73.00 | 17.4 |
| Rio de Janeiro | −94.22 | 45.1 | −75.00 | 25.6 |

It can be recognized that as the number of daily infections has decreased there has been a recovery in the use of urban transport and private vehicles. Since urban transport is managed by the government, it is observed that its recovery happens slower than that of private vehicles, which is identified as the recovery of congestion. TCI has a higher correlation with infections than the use of urban transport.

Figure 10 shows for the months of February, April, and June the number of people or passengers using the BRT Transmilenio urban transport system in the city of Bogotá for the years 2017 to 2020, which, in this city, is the urban transport system of reference consisting of exclusive transit buses. The figure shows that the number of people using the system fell below 20,000,000 per day in April 2020, which is down 18%, compared to the same month in previous years. Additionally, there was a slight recovery in the number of passengers for the month of June 2020, but they did not exceed 25% of the same month in previous years [48]. According to "it can be argued that these findings do not support the effectiveness of suspending mass urban transport systems as a pandemic countermeasure aimed at reducing or slowing population spread because, whatever the relevance of public transport is to individual-level risk, household exposure most likely poses a greater threat" [48].

Interestingly, the point of greatest decline in the use of private vehicles was greater than that of public transport. However, in each city, the speed of recovery of the mobility of particular vehicles was greater than that of public transport. This can be due to two factors: public transport follows the guidelines dictated by the regional or national government; it is also possible that due to fear of infection, people use private vehicles instead of public transport. Likewise, the rebound in use was earlier for private vehicles than for public transport.

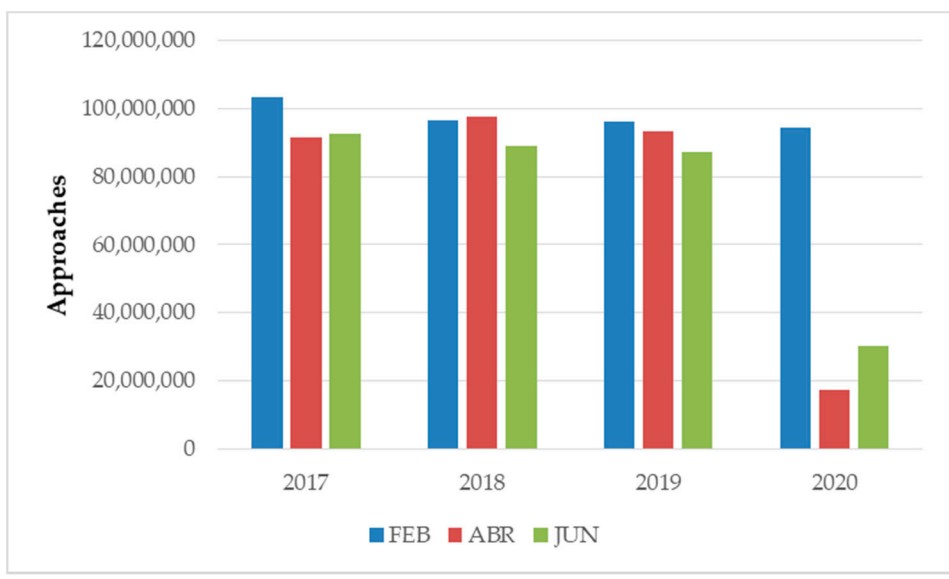

**Figure 10.** Validations in the BRT Transmilenio (Bogotá, Colombia) in the months of February, April and June, from the years 2017 to 2020.

An indicator of the change in the behavior model associated with mobility restrictions due to the pandemic is the increase in internet data traffic [49], affecting some cities, such as Bogotá, up to 40%. Another indicator is the behavior of people, as shown in Table 4, depending on the destination of their movements during the months of the COVID-19 pandemic. In this study, it is observed that trips to second homes only increased in the five countries considered in the study. In turn, there was a significant reduction in visits to parks, this being the main destination affected by the reduction in mobility in Argentina, Brazil and Chile, while in Peru and Colombia, respectively, the destination to shops and leisure, and to transport stations were reduced. The greatest average decrease in trips, regardless of destination, occurred in Chile, with an average decrease of −59%, followed by Perú and Argentina, both with decreases of over −50%. On the contrary, the lowest average decrease in trips was observed in Brazil, with a value of −28%, highlighting in this country the destinations of supermarkets and pharmacies, which practically reached the same values as those before the start of the pandemic. Table 4 shows in bold the maximum drop in the mobility percentage according to the destination for each country. The cities of Rio de Janeiro, Mexico City and São Paulo are ranked 2, 4 and 5, respectively, among the most populated cities in the world [32], which demonstrates the representativeness of the reduction of destinations.

**Table 4.** Percentage of variation (increase +, and reduction −) of mobility, according to destinations, during the COVID-19 pandemic in Argentina, Brazil, Chile, Colombia and Perú.

| Percentage of Variation of Mobility According to Destinations (%) | Country | | | | |
| --- | --- | --- | --- | --- | --- |
| | Argentina | Brazil | Chile | Colombia | Perú |
| Shops and Leisure | −61 | −40 | −66 | −47 | **−67** |
| Supermarkets and Drugstores | −21 | −2 | −46 | −24 | −35 |
| Parks | **−83** | **−39** | **−67** | −38 | −46 |
| Transport stations | −54 | −38 | −66 | **−48** | −58 |
| Work | −28 | −22 | −50 | −35 | −54 |

## 5. Conclusions

This study analyzes the impact of the COVID-19 pandemic on motorized mobility in large megacities belonging to five of the most populated countries in Latin America, including Bogotá, Buenos Aires, Mexico City, Santiago, Lima, Brasilia, São Paulo, and Rio de Janeiro. For this, the public data made available to society by different public

and private institutions, such as the Inter-American Development Bank, Johns Hopkins Hospital University, and the Moovit© platform, were used. These institutions have made the data public in order to better understand the contagion process of the pandemic, its impacts, and the effectiveness of the health control measures implemented.

For each city, a descriptive analysis of the recovery of mobility from traffic congestion and urban public transport is made, depending on the level of contagion. As expected, the recovery of traffic congestion occurs earlier and faster than that of urban transport, since the latter depends on the control measures imposed in each locality, highlighting that the decrease in the number of infections brings the flexibility of social distancing measures and the recovery of vehicular mobility, which is observed as a function of the increase in MRR. Only the city of Buenos Aires does not show a clear recovery in cases of contagion by COVID-19, unlike the other cities analyzed. However, with adequate sanitation measures, public transport is more efficient for reducing the risks of contagion.

In the work, a decoupling is observed between the evolution of the number of COVID-19 cases and the recovery rate of urban public transport, which is explained by the perception of the risk of the population being infected in this type of transport. Certainly, according to [50], the following factors affect the risk of contagion in public transport systems and must be considered to guarantee the mobility of citizens with the highest occupancy levels: (a) user behavior related to masks, eye protection, and quiet travel to reduce airborne transmission, (b) the type of ventilation system of the vehicle and frequency of air renewal, (c) proximity of drivers, (d) duration of the trip and, finally, (e) cleaning and disinfection of high-contact surfaces.

The main recommendations of the World Health Organization (WHO) [51] for the transformation of transport in cities establish lines of concrete actions for its future in cities, among which are the adoption of clean methods of electricity generation; the prioritization of rapid urban transport, pedestrian and bicycle paths in cities, interurban transport of cargo and passengers by rail; and the use of cleaner diesel-engine heavy-duty vehicles and low emission vehicles and fuels, especially with low sulfur fuels.

The pandemic and the different mobility restrictions have highlighted the need for more agile and less collective forms of transport services, representing an opportunity for the development of new public–private transport services that would contribute to sustainability, resilience, mitigation of climate change, and the health of the population. In the case of Latin America and the studied cities, several challenges must be addressed, such as the quality of public transport, road safety for the most vulnerable (pedestrians), emissions and their impact on health, as well as the high traffic congestion and optimization of road infrastructure and finally, the effective prioritization of buses, cyclists and pedestrians.

The COVID-19 pandemic has revealed less collective and more agile forms of mobility, which is an important opportunity for the region to develop new forms of transport, such as soft and personal forms of mobility which were observed as the link between the evolution of the number of COVID-19 cases and the recovery rate of urban public transport, which is explained by the perception of the risk of the population being infected in this type of transport.

**Author Contributions:** Conceptualization, R.A., J.O.-O., R.R.-P., L.M.N.-G. and C.L.V.; methodology, R.A., J.O.-O., C.L.V., L.M.N.-G., R.R.-P. and M.I.G.-C.; validation, R.A., R.R.-P. and M.G.-A.; formal analysis, J.O.-O., C.L.V., L.M.N.-G., R.R.-P., M.I.G.-C. and M.G.-A.; investigation, J.O.-O., C.L.V. and M.G.-A.; data curation, R.A., L.M.N.-G. and R.R.-P.; writing—original draft preparation, C.L.V., L.M.N.-G., R.R.-P., M.I.G.-C. and M.G.-A.; writing—review and editing, R.A., J.O.-O., M.I.G.-C. and C.L.V.; visualization, R.A., J.O.-O. and L.M.N.-G.; supervision, R.R.-P., C.L.V., M.I.G.-C. and M.G.-A.; project administration, L.M.N.-G., R.R.-P. and C.L.V. All authors have read and agreed to the published version of the manuscript.

**Funding:** This research was funded by the Sustainable Urban Transport and Mobility Network (RITMUS, 718RT0566) of the Science and Technology for the Development of Ibero-America Program (CYTED), Carlemany University and University Foundation "Los Libertadores".

**Institutional Review Board Statement:** This study did not involve humans or animals.

**Informed Consent Statement:** This study did not involve humans.

**Data Availability Statement:** This study did not report any data.

**Acknowledgments:** The authors of this article would like to thank the Science and Technology for the Development of Ibero-America Program (CYTED), since it was developed within the framework of the Sustainable Urban Transport and Mobility Network (RITMUS, 718RT0566). Additionally, the authors would also like to thank University Foundation "Los Libertadores" and Carlemany University for supporting the article.

**Conflicts of Interest:** The authors declare no conflict of interest.

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
