# Peer review of "Behavior of Traffic Congestion and Public Transport in Eight Large Cities in Latin America during the COVID-19 Pandemic"

_applsci, doi:10.3390/app11104703_

Round 1
Reviewer 1 Report
the subject matter is topical. Figures 1 to 7 should have the title of the axis at the top and not in the middle for better reading. It is necessary to include in the introductory part more literature references related to COVID-19 and the modification of modal choices starting from the first affected states like China and Italy in Europe. Therefore we recommend reading the following works : 1) Charoenwong, B., Kwan, A., & Pursiainen, V. (2020). Social connections with COVID-19–affected areas increase compliance with mobility restrictions. Science advances, 6(47), eabc3054. 2) Mok, K. H., Xiong, W., Ke, G., & Cheung, J. O. W. (2021). Impact of COVID-19 pandemic on international higher education and student mobility: Student perspectives from mainland China and Hong Kong. International Journal of Educational Research, 105, 101718. 3) Moslem, S., Campisi, T., Szmelter-Jarosz, A., Duleba, S., Nahiduzzaman, K. M., & Tesoriere, G. (2020). Best–worst method for modelling mobility choice after COVID-19: evidence from Italy. Sustainability, 12(17), 6824. 4) Campisi, T., Basbas, S., Skoufas, A., Akgün, N., Ticali, D., & Tesoriere, G. (2020). The Impact of COVID-19 Pandemic on the Resilience of Sustainable Mobility in Sicily. Sustainability, 12(21), 8829. 5) Scorrano, M., & Danielis, R. (2021). Active mobility in an Italian city: Mode choice determinants and attitudes before and during the Covid-19 emergency. Research in Transportation Economics, 101031. It should mention all available forms of mobility in the examined contexts and whether e.g. shared mobility or taxis have also decreased in use. Likewise for short trips the use of bicycles or walking or micromobility. In this way the before and after view of the transport system related to the offer of the examined areas becomes clear. The purpose of this work and the methodology used to compare the data should be defined in the introduction and the choice of cities should be justified. A more detailed explanation should be included to accompany Figure 9. Why are the data in Table 3 in two colours?Author Response
Figures 1 to 7 should have the title of the axis at the top and not in the middle for better reading |
The position of the axis titles has been modified according to the reviewer's suggestion. The modified figures are from Figure 2 to Figure 9. |
It is necessary to include in the introductory part more literature references related to COVID-19 and the modification of modal choices starting from the first affected states like China and Italy in Europe. |
The Introduction section has been extended by adding more references on the impact of COVID-19 on changes in mobility behaviour, with special reference to the two main global foci. These new references are: [2], [3], [4], [6], [7], [8], [9], [10], [11] , [12], [13] y [14]. |
Therefore we recommend reading the following works : |
|
1) Charoenwong, B., Kwan, A., & Pursiainen, V. (2020). Social connections with COVID-19–affected areas increase compliance with mobility restrictions. Science advances, 6(47), eabc3054. |
Incorporated as reference [2]. |
2) Mok, K. H., Xiong, W., Ke, G., & Cheung, J. O. W. (2021). Impact of COVID-19 pandemic on international higher education and student mobility: Student perspectives from mainland China and Hong Kong. International Journal of Educational Research, 105, 101718. |
Incorporated as reference [12]. |
3) Moslem, S., Campisi, T., Szmelter-Jarosz, A., Duleba, S., Nahiduzzaman, K. M., & Tesoriere, G. (2020). Best–worst method for modelling mobility choice after COVID-19: evidence from Italy. Sustainability, 12(17), 6824. |
Incorporated as reference [9]. |
4) Campisi, T., Basbas, S., Skoufas, A., Akgün, N., Ticali, D., & Tesoriere, G. (2020). The Impact of COVID-19 Pandemic on the Resilience of Sustainable Mobility in Sicily. Sustainability, 12(21), 8829. |
Incorporated as reference [6]. |
5) Scorrano, M., & Danielis, R. (2021). Active mobility in an Italian city: Mode choice determinants and attitudes before and during the Covid-19 emergency. Research in Transportation Economics, 101031. |
Incorporated as reference [3]. |
It should mention all available forms of mobility in the examined contexts and whether e.g. shared mobility or taxis have also decreased in use. Likewise, for short trips the use of bicycles or walking or micromobility. |
The implications of the COVID-19 pandemic on other forms of urban mobility have been included in the Discussion section, from line 231 to line 240. |
In this way the before and after view of the transport system related to the offer of the examined areas becomes clear |
|
The purpose of this work and the methodology used to compare the data should be defined in the introduction and the choice of cities should be justified. |
In the Introduction section, a paragraph has been included (from line 102 to line 114) explaining the methodology used to compare the data and the justification for the choice of cities. |
A more detailed explanation should be included to accompany Figure 9. |
The explanation of Figure 9 (now Figure 10) has been included in the paragraph from line 266 to line 276. |
Why are the data in Table 3 in two colours? |
The explanation of the data in red in Table 3 (now Table 4) is included in the paragraph from line 297 to 301. |
Reviewer 2 Report
The authors made an assessment of mobility aspects in several cities from Latin America. Although the data presented in interesting, it would be better if:
- the authors also represent the data on a geographical map, so people that are not that familiar to Latin America can understand the spread of the effects presented
- the authors also present the population in those cities compared to the total population of the country (to see the relevance of the data in Table 3)
- the authors should extend the period analyzed, comparing the data with the previous year (as I understand from the methodology, the reduction is reported to the period March 1-7 2020, but how can you compare a month of July (e.g.) with March? The traffic is not constant all year long...)
- it is important that the authors split the effect of the mobility reduction into two aspects: restrictions from the authorities (Covid quarantine), and voluntary restriction of movement, without enforcement from the authorities.
Author Response
The authors also represent the data on a geographical map, so people that are not that familiar to Latin America can understand the spread of the effects presented |
The requested map is included in Figure 1. |
|
The authors also present the population in those cities compared to the total population of the country (to see the relevance of the data in Table 3) |
Comparative information on the population of the cities studied and that of their respective countries has been included in Table 1. |
|
The authors should extend the period analyzed, comparing the data with the previous year (as I understand from the methodology, the reduction is reported to the period March 1-7 2020, but how can you compare a month of July (e.g.) with March? The traffic is not constant all year long...) |
The extension of the period analysed to the year before the start of the COVID-19 pandemic has been included in the Discussion section, from line 222 to line 253. |
|
It is important that the authors split the effect of the mobility reduction into two aspects: restrictions from the authorities (Covid quarantine), and voluntary restriction of movement, without enforcement from the authorities. |
The available data do not allow to discriminate between the two aspects mentioned by the reviewer, so this correction could not be addressed. |
|
Round 2
Reviewer 1 Report
There are errors related to axis of figures on the results paragraph. Please put the data in the axis above and not in the middle of figures.
There are some errors in the text.
After this review the paper Wil be ready for pubblication
Author Response
COMMENTS OF THE REVIEWERS | COMMENTS OF AUTHORS TO THE REVIEWERS |
REVIEWER 1 - ROUND 2 | |
There are errors related to axis of figures on the results paragraph. | The Results paragraph has been amended, from line 201 to line 205, as recommended by the reviewer. |
Please put the data in the axis above and not in the middle of figures. | The axes of the figures from Figure 2 to Figure 9 have been changed according to the reviewer's recommendation. |
There are some errors in the text. | The authors have reviewed the article throughout the review process and have corrected all the errors we have detected. If Reviewer 1 identifies further errors in the text, please let us know what they are, so that we can correct them. |
Reviewer 2 Report
The authors have addressed all the observations made in the first stage of the review, except one: "It is important that the authors split the effect of the mobility reduction into two aspects: restrictions from the authorities (Covid quarantine), and voluntary restriction of movement, without enforcement from the authorities.". The authors replied that there isn't enough data to do this. I think that there are some information publicly available regarding quarantine measures or different movement restrictions, as already mentioned in the paper (226-228).
It is obvious that an extensive research in this direction will exceed the purpose of this paper. But I still consider that a minor research regarding these aspects is useful.